# *ADRA2A* and *IRX1* are putative risk genes for Raynaud's phenomenon

Sylvia Hartmann[1], Summaira Yasmeen [1], Benjamin M. Jacobs [2], Spiros Denaxas[3,4,5,6], Munir Pirmohamed [7], Eric R. Gamazon [8], Mark J. Caulfield [9], Genes & Health Research Team*, Harry Hemingway [3,4,6], Maik Pietzner [1,10,11,12] ✉ & Claudia Langenberg [1,10,11,12] ✉

Raynaud's phenomenon (RP) is a common vasospastic disorder that causes severe pain and ulcers, but despite its high reported heritability, no causal genes have been robustly identified. We conducted a genome-wide association study including 5,147 RP cases and 439,294 controls, based on diagnoses from electronic health records, and identified three unreported genomic regions associated with the risk of RP ($p < 5 \times 10^{-8}$). We prioritized *ADRA2A* (rs7090046, odds ratio (OR) per allele: 1.26; 95%-CI: 1.20-1.31; $p < 9.6 \times 10^{-27}$) and *IRX1* (rs12653958, OR: 1.17; 95%-CI: 1.12–1.22, $p < 4.8 \times 10^{-13}$) as candidate causal genes through integration of gene expression in disease relevant tissues. We further identified a likely causal detrimental effect of low fasting glucose levels on RP risk ($r_G = -0.21$; p-value = $2.3 \times 10^{-3}$), and systematically highlighted drug repurposing opportunities, like the antidepressant mirtazapine. Our results provide the first robust evidence for a strong genetic contribution to RP and highlight a so far underrated role of $\alpha_{2A}$-adrenoreceptor signalling, encoded at *ADRA2A*, as a possible mechanism for hypersensitivity to catecholamine-induced vasospasms.

Raynaud's phenomenon (RP) is a common episodic, vasospastic disorder that affects 2–5% of the population and can severely affect an individual's quality of life by causing pain or even ulcers[1–3]. RP typically manifests with bi- or triphasic colour change in fingers and toes because of vasospasms in arteriovenous anastomoses responsible for thermoregulation, which can be triggered by cold or emotional stress[4]. The causes for the more common form of primary RP are largely unknown[4,5], while secondary RP is diagnosed as a consequence of connective tissue diseases, like systemic lupus erythematosus (SLE) or

systemic sclerosis (SSc), or triggered by the use of drugs such as beta-blockers. Management of RP is predominantly limited to the avoidance of triggers and evidence for medical treatment is generally weak. Repurposed vasodilators are the first line of treatment if pharmacological intervention is required due to progressive frequency of vasospastic attacks[6], although only calcium channel blockers have so far been shown to lead to a significant and reproducible reduction in the frequency of vasospastic attacks[7]. However, the use of systemic drugs for localized symptoms puts patients at risk of generalized

[1]Computational Medicine, Berlin Institute of Health at Charité—Universitätsmedizin Berlin, Berlin, Germany. [2]Preventive Neurology Unit, Wolfson Institute of Population Health, Queen Mary University of London, London, UK. [3]Institute of Health Informatics, University College London, London, UK. [4]Health Data Research UK, London, UK. [5]British Heart Foundation Data Science Centre, London, UK. [6]National Institute of Health Research University College London Hospitals Biomedical Research Centre, London, UK. [7]Department of Pharmacology and Therapeutics, The Wolfson Centre for Personalised Medicine, University Liverpool, Liverpool, UK. [8]Division of Genetic Medicine and Vanderbilt Genetics Institute, Vanderbilt University Medical Center, Nashville, TN 37232, USA. [9]William Harvey Research Institute, Queen Mary University of London, London, UK. [10]MRC Epidemiology Unit, University of Cambridge, Cambridge, UK. [11]Precision Healthcare University Research Institute, Queen Mary University of London, London, UK. [12]These authors jointly supervised this work: Maik Pietzner, Claudia Langenberg. *A list of authors and their affiliations appears at the end of the paper. ✉e-mail: maik.pietzner@bih-charite.de; claudia.langenberg@bih-charite.de

adverse effects, such as hypotension. While there are now trials investigating local application of botulinum toxin to mitigate vasospastic effects with early promising results[8], a better understanding of the underlying mechanisms is needed to develop safe and effective treatments.

RP is highly heritable with estimates of 55–64% being reported[9,10], but previous candidate gene studies[11,12], like at serotonin receptors[13], and an early, small ($n = 640$ cases) genome-wide association study (GWAS)[14] failed to provide evidence for any robustly associated regions or genes. In-depth investigation and integration of detailed information from primary and secondary healthcare records with genetic array data now provide the opportunity to study under-investigated diseases with diagnostic specificity at an unprecedented scale.

Here, we present the so far largest GWAS for RP including 5147 cases in the UK Biobank cohort[15] and report two robust and strong ($p < 4.8 \times 10^{-13}$) novel loci. We highlight two independent disease mechanisms supported by those loci that challenge and advance our current understanding of primary RP, with *ADRA2A* highlighting the role of $\alpha_{2A}$-adrenoreceptors and *IRX1* as a putative regulator of vasodilation by altering prostaglandin and/or bradykinin responsiveness.

## Results

We identified a total of 5147 RP cases and 439,294 controls of European descent included in the genetic analyses (Supplementary Table 1 and Supplementary Fig. 1) based on collation and evidence of absence or presence of diagnostic codes from electronic health records (ICD-10 codes: I73.0, I73.00, I73.01; CTV3/Read2: G730., G7300, G7301, G730z, XE0VQ), including 2185 prevalent cases and 2962 incident cases. We followed the recommendation by Wigley et al.[4] to summarize all patients with a relevant code under the term Raynaud's phenomenon (RP) rather than Raynaud's syndrome.

### Genome-wide association analysis

We identified a total of three genome-wide significant novel loci ($p < 5.0 \times 10^{-8}$, minor allele frequency (MAF) 14.3–31.1%) associated with RP (Fig. 1 and Table 1; regional association plots in Supplementary Fig. 2). Two of which met a more stringent significance threshold ($p < 1.0 \times 10^{-9}$; Table 1), which we consider as tier 1 findings of which one had previously been shown to increase the risk of myocardial infarction (rs12653958). All lead variants increased the risk for RP by ~20% per copy of the minor allele (Table 1) and resided in intergenic

regions with no obvious functional variants in linkage disequilibrium (LD; $r^2 > 0.6$) and were of low predicted impact (CADD score range: 0.36–3.41). Despite the established sex-difference of RP, we did not find evidence that the effects of any of the identified lead signals differed between men and women (all $p$-values for interaction > 0.05; Supplementary Table 2). We estimated a single nucleotide polymorphism (SNP)-based heritability on the liability scale of 7.7% (95% CI: 4.8–10.6%, $p < 2.4 \times 10^{-7}$) with little evidence of genomic inflation (LD-score intercept 1.01; Supplementary Fig. 3).

To evaluate whether genetic findings were specific to the development of RP or driven by associations with diseases leading to secondary RP, we performed sensitivity analyses based on a more stringent definition of primary RP (3505 cases, 68.1%) that excluded potential secondary cases (see the "Methods" section). Effect estimates for both tier 1 loci were highly consistent in size and direction (Supplementary Table 3) and remained genome-wide significant ($p < 1.4 \times 10^{-12}$ for both), indicating their potential role in the idiopathic origin of primary RP, whereas the association within the MHC region was attenuated.

### *ADRA2A* and *BBIP1* are likely effector genes at 10q25.2

The lead variant at this strongest locus, rs7090046, is located 24.1k bp downstream to *ADRA2A* and 181.9k bases downstream from *BBIP1* (Fig. 2) and Bayesian fine-mapping identified a 95%-credible set of only four variants (see the "Methods" section), comprising rs7090046, rs1343449, rs1343451, and rs7084501 with posterior inclusion probabilities (PIP) of 48.6%, 23.6%, 12.5% and 12.6% of the GWAS signal, respectively. Variant rs1343449 is a reported gene expression quantitative trait locus (cis-eQTL) for *ADRA2A* in artery tissue and statistical colocalization confirmed a shared genetic signal with RP with high confidence (tibial artery, posterior probability (PP) > 96%; Fig. 2), with the RP risk-increasing A-allele of rs7090046 being associated with higher *ADRA2A* expression (beta = 0.28, $p = 1.0 \times 10^{-9}$). We also observed evidence of colocalization with *BBIP1* expression in the brain hippocampus and colon (PP > 98%; Fig. 2 and Supplementary Data 1), with the A-allele being associated with lower *BBIP1* expression in brain hippocampus (beta = −0.37, $p = 9.1 \times 10^{-9}$), raising the possibility that variants in the credible set have pleiotropic, tissue-dependent effects in different directions.

*ADRA2A* encodes $\alpha_{2A}$-adrenoreceptors, responsible for mediating response to stress in the central and peripheral autonomous nervous system[16], and vasoconstriction in response to catecholamine release in

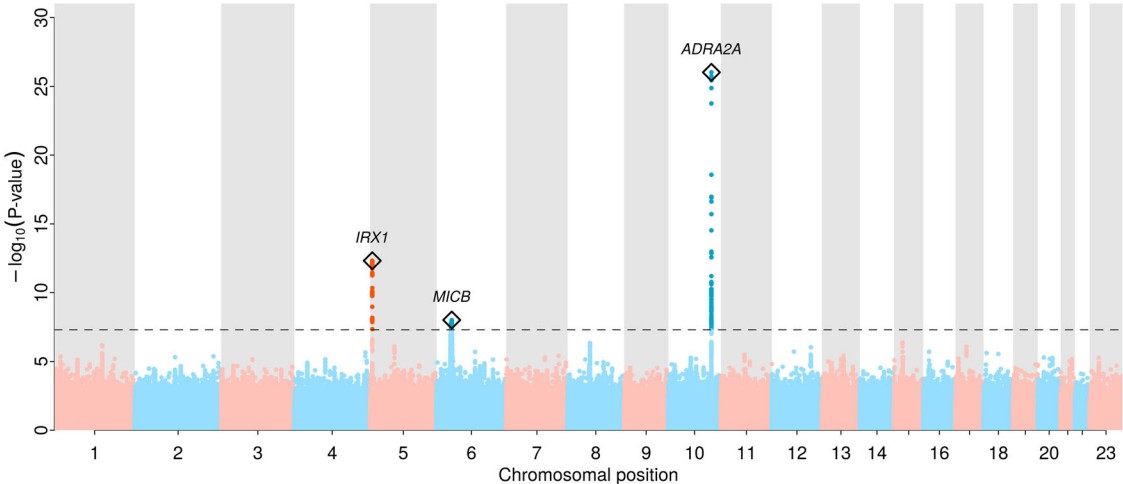

**Fig. 1 | Manhattan plot of genome-wide association results for Raynaud's phenomenon (5147 RP cases and 439,294 controls).** Distribution of RP-associated single nucleotide polymorphisms across the genome, −log10(P-values) from logistic regression models are plotted for each variant. The grey dashed line indicates the genome-wide significant threshold at $p = 5 \times 10^{-8}$. Regional sentinel variants at significant loci are highlighted with a diamond and the closest gene, if any, is annotated.

**Table 1 | Regional lead variants for the eight loci significantly associated with Raynaud's phenomenon**

| Tier | SNP | Chr | Position | Alleles | MAF | OR (95% CI) | P-value | Closest gene[a] |
|---|---|---|---|---|---|---|---|---|
| 1 | rs7090046 | 10 | 112860930 | A/G | 0.31 | 1.26 (1.20, 1.31) | $9.58 \times 10^{-27}$ | *ADRA2A* |
| 1 | rs12653958 | 5 | 4032849 | A/G | 0.30 | 1.16 (1.12, 1.22) | $4.76 \times 10^{-13}$ | *IRX1* |
| 2 | rs3094013 | 6 | 31434366 | G/A | 0.14 | 1.17 (1.11, 1.24) | $9.73 \times 10^{-9}$ | *MICB* |

Tier = categorization whether genetic loci met a stringent Bonferroni threshold for significance (tier 1; $p < 1.0 \times 10^{-9}$) or the standard genome-wide significance threshold (tier 2; $p < 5.0 \times 10^{-8}$).
$N = 444{,}441$, 5147 cases, and 439,294 controls. Alleles are given as effect allele/non-effect allele.
*Chr* chromosome, *MAF* minor allele frequency, *OR* odds ratio from logistic regression models.
[a]Protein coding gene in italics.

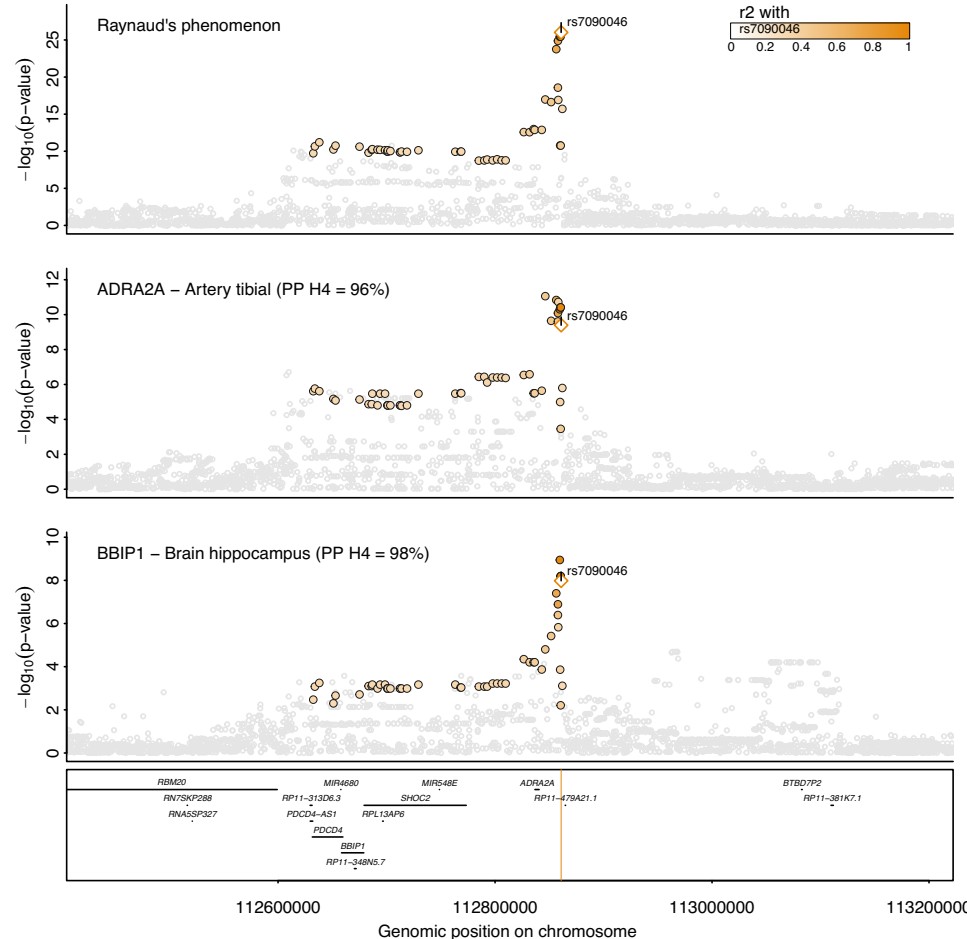

**Fig. 2 | Regional association plot at *ADRA2A/BBIP1*.** Regional association plot for Raynaud's phenomenon (top), gene expression of *ADRA2A* in a tibial artery (middle), and gene expression of *BBIP1* in the hippocampus (bottom). *ADRA2A* and *BBIP1* have been prioritized as candidate causal genes based on statistical colocalization. Summary statistics from logistic regression models for RP are from the present study whereas summary statistics from linear regression models were obtained from GTEx v8[22]. Colouring of SNPs is based on linkage disequilibrium with the lead RP variant (rs7090046) at this locus. Numbers in brackets indicate posterior probabilities (PP) for a shared genetic signal with RP based on statistical colocalization.

the small arteries and arterioles[17]. Our finding may imply that overstimulation or increased expression of $\alpha_{2A}$-adrenergic receptors contributes to the vasospastic effects characteristic of RP and its symptoms, also in line with RP being an adverse effect of $\alpha_2$-adrenergic agonists such as clonidine[4,18,19]. A possible role of *BBIP1*, in contrast, is less obvious. *BBIP1* encodes for a protein in the BBSome complex, which is involved in the signalling from and to cilia.

Investigating thousands of other outcomes from the GWAS catalogue[20] and the Open Targets platform[21] in addition to phenome-wide analyses of UK Biobank ($p < 10^{-6}$, see the "Methods" section), we observed no other significant associations for rs7090046 or its proxies ($r^2 > 0.6$), except for "specified peripheral vascular disease" in UK Biobank (OR 1.18, $p$-value $4.5 \times 10^{-11}$), a non-specific code that comprises, among other diseases, RP. These results suggest the specificity of the mechanism underlying the genetic signal at 10q25.2 on RP.

### *IRX1* is the likely effector gene at 5p15.33 and links to cardio-vascular outcomes

We fine-mapped the second strongest signal at 5p15.33 to 24 variants with rs12653958 as the lead SNP with a PIP of 6.4%. The credible set also covered rs11748327 (PIP = 4.7%) which has been reported as a cis-eQTL for *IRX1* in skeletal muscle[22]. Statistical colocalization further confirmed a shared genetic signal between RP and *IRX1* expression in skeletal muscle and tibial artery with high (PP = 99%) and moderate (PP = 47%) confidence, respectively (Fig. 3). The RP-increasing effect-

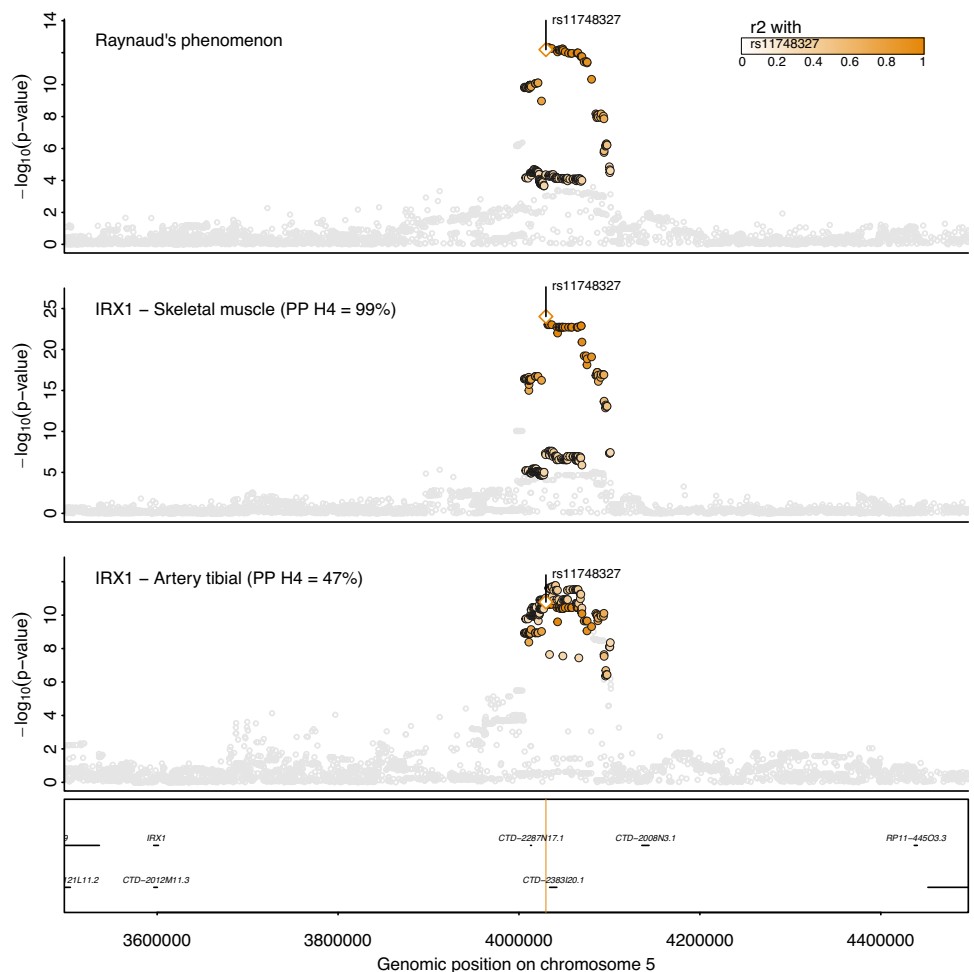

**Fig. 3 | Regional association plot at *IRX1*.** Regional association plot for Raynaud's phenomenon (top), gene expression of IRX1 in skeletal muscle (middle), and gene expression of IRX1 in tibial artery (bottom). Summary statistics from logistic regression models for RP are from the present study whereas summary statistics from linear regression models were obtained from GTEx v8[22]. Colouring of SNPs is based on linkage disequilibrium with the lead RP variant (rs12653958) at this locus. Numbers in brackets indicate posterior probabilities (PP) for a shared genetic signal with RP based on statistical colocalization.

allele of rs11748327 was associated with higher expression of *IRX1* in skeletal muscle (beta = 0.57, $p = 1.1 \times 10^{-22}$) and tibial artery (beta = 0.42, $p = 5.7 \times 10^{-11}$), suggesting that higher gene expression confers a higher risk for RP. *IRX1* encodes for members of the homeobox-containing genes that have been shown to be involved in embryonic development and cellular differentiation[23]. The lead variant, rs12653958, at this locus, tags a cluster of variants ($r^2 = 0.7–1.0$) that has previously been reported to be associated with myocardial infarction (MI)[24,25] and coronary artery disease (CAD)[26] among East Asian populations and is further a cross-ancestral signal for red blood cell phenotypes[27,28]. In detail, the RP risk-increasing G-allele was also associated with an increased risk for coronary artery disease (beta = −0.06; $p < 1.8 \times 10^{-8}$) in data from BioBank Japan[26]. The fact that this common variant ($MAF_{EUR} = 30.0\%$; $MAF_{EAS} = 22.3\%$) has not yet been identified for MI or CAD in Europeans ($p = 1.61 \times 10^{-7}$) despite much larger sample sizes ($n = 181{,}522$)[29], might point to the possibility of a yet to be identified gene–environment interaction or other characteristics distinct to the East Asian cohorts that account for this observation.

## RP in individuals of South Asian ancestry
We aimed to test for replication and transferability of our European-centric results in 401 RP cases of British Bangladeshi and Pakistani ancestry from the Genes & Health cohort[30] ($n = 40{,}532$), and observed directionally concordant results (rs11748327; OR = 1.16, 95%-CI = 0.99–1.37, *p*-value = 0.06; *IRX1*) for one of the two strongest loci,

while for the second one, rs7090046 (*ADRA2A*), the lead variant or suitable genetic proxies were not available. However, further studies with comparable numbers of RP cases are needed to establish trans-ethnic and ethnic-specific risk variants for RP.

## Genetic correlation between RP and other traits
We next tested for a shared genetic architecture (genetic correlation: $r_G$) between RP and known or suspected risk factors and comorbidities with a vascular or immune origin/component like peripheral artery disease (PAD) or COVID-19 (Supplementary Data 2). We observed that possible causes or related disorders for RP, like migraine ($r_G = 0.21$, $p = 9.0 \times 10^{-3}$), PAD ($r_G = 0.38$, $p = 2.1 \times 10^{-3}$), or SLE ($r_G = 0.33$ $p = 1.0 \times 10^{-3}$) significantly (false discovery rate < 5%) correlated when considering all RP patients, but were strongly attenuated once we computed genetic correlations restricted to patients with primary RP (Supplementary Fig. 5, Supplementary Data 3). In contrast, other significant correlations persisted ($p < 0.05$) when we restricted genetic correlation analysis to results based on patients with primary RP (to which we refer to as *robust* in the following). This included significant inverse correlations with type 2 diabetes ($r_G = -0.14$, $p = 2.7 \times 10^{-3}$), fasting glucose ($r_G = -0.21$, $p = 0$, $p = 2.3 \times 10^{-3}$), and a positive correlation with HDL-cholesterol levels ($r_G = 0.13$, $p = 9.2 \times 10^{-4}$) (Supplementary Fig. 5).

To distinguish whether genetic correlations represent causal directions from versus towards RP or are due to shared risk factors that

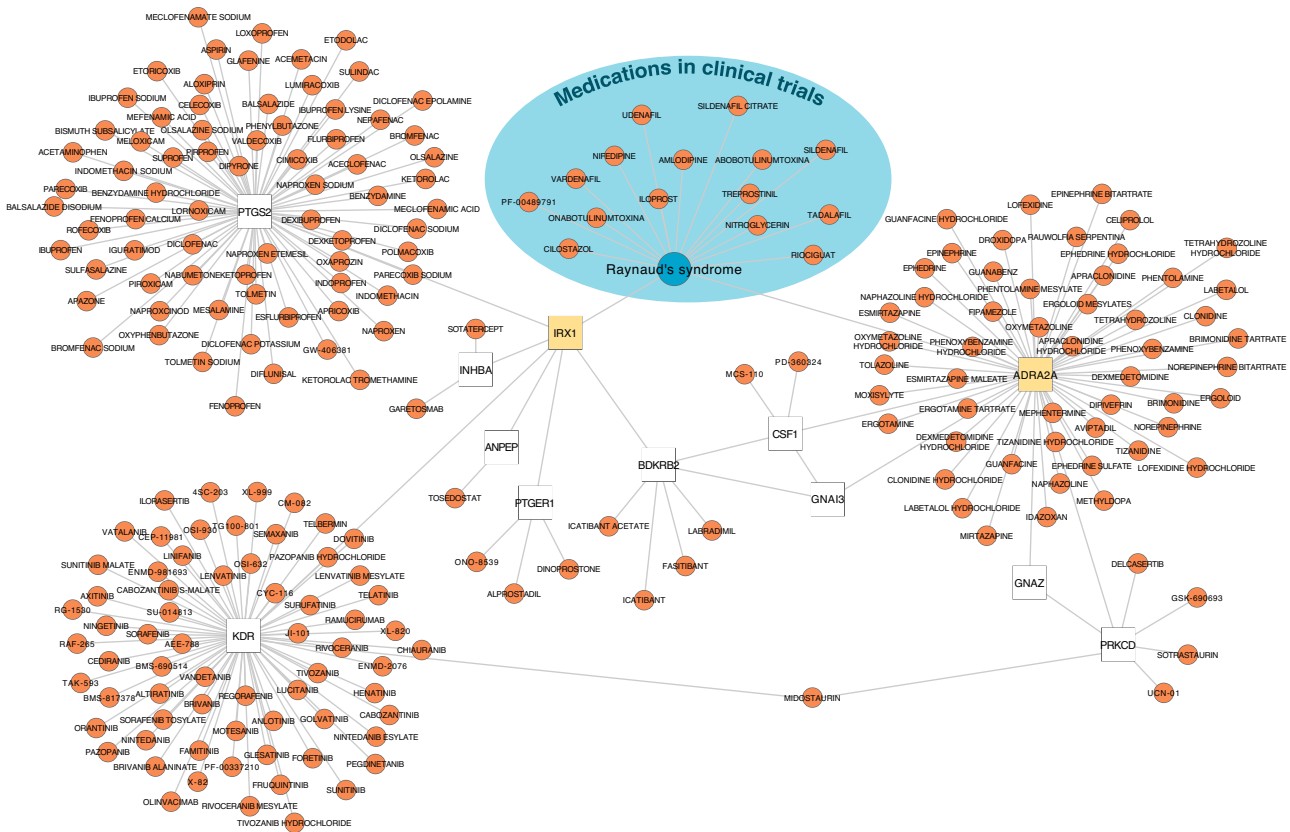

**Fig. 4 | Gene–Drug–Disease network for Raynaud's phenomenon.** The network displays drug target information obtained from Open Targets[21] for (1) Raynaud's Phenomenon (RP; blueish area), (2) genes associated with RP in the present study (yellow), and (3) interacting genes (white) based on functional work[32].

likely act independently on RP and correlated traits, we performed latent causal variable analysis[31] (see the "Methods" section). Out of a total of six phenotypes with evidence for significant genetic correlations, only SLE (genetic causality proportion (GCP) = 0.75; $p < 1.7 \times 10^{-4}$) and fasting glucose (GCP = 0.77; $p < 5.21 \times 10^{-15}$) showed strong evidence for a causal genetic effect on RP risk, of which fasting glucose but not SLE persisted in analyses considering only patients with primary RP (GCP = 0.80; $p < 1.3 \times 10^{-16}$). These results indicate that low fasting glucose levels may causally increase the risk for RP, while frequent comorbidities of RP, like migraine, are likely explained by shared risk factors.

We then extended the targeted analysis by computing genetic correlations with 185 medical concepts—'phecodes', to gain a more comprehensive view of possible diseases that share a genetic architecture with RP (see the "Methods" section; Supplementary Data 4). We observed 8 phecodes with significant, although moderate, genetic correlations with RP (Supplementary Data 4), of which only the significant genetic correlation with osteoporosis persisted when we restricted the analysis to patients with primary RP ($r_G = 0.35$, $p < 8.7 \times 10^{-5}$). However, none of the phenotypes showed evidence of a direct causal link towards or from RP in latent causal variant analysis (see the "Methods" section; Supplementary Table 4).

**From genes to druggable targets**
We finally searched for putative druggable targets for RP based on the identified loci by cross-referencing high-confidence gene assignments and functional interaction partners thereof[32] in the Open Targets database[21] (see the "Methods" section). While we identified no link for drugs with known (genetic) targets currently or previously in clinical trials for RP (e.g., phosphodiesterase inhibitors), we identified a total of 211 drugs that target either directly (e.g., *ADRA2A*) or indirectly (e.g., *PTGS2*) gene products identified in our

study (Fig. 4 and Supplementary Data 5). More than 50 drugs had at least some affinity to $\alpha_{2A}$-adrenergic receptors, although none did so specifically. Since we identified increased expression of $\alpha_{2A}$-adrenergic receptors as a possible disease-causing mechanism, partially selective antagonists, such as the widely used antidepressant mirtazapine, might be promising candidates for repurposing. Among the drugs indirectly linked to RP-associated gene products, those counteracting vasoconstriction or promoting vasodilation have the greatest repurposing potential. For example, drugs promoting prostaglandin (via *PTGER1* but not *PTGS2*) or bradykinin signalling (via *BDKRB2*) such as dinoprostone or labradimil, respectively, might exert beneficial effects. However, the indirect nature of the gene–drug–disease link for the latter (via altered *IRX1* expression) requires further functional follow-up to establish the precise underlying mechanism and effect directions.

## Discussion
RP is a common but understudied disease with a high heritable component and identifying responsible genes can advance our understanding of the aetiology and eventually guide the development of treatment strategies. We identified, for the first time, two robust susceptibility loci. Through integration with gene expression data, we demonstrated that the strongest locus likely acts via increased expression of $\alpha_{2A}$-adrenoreceptors in arterial tissue, augmenting the current concept of predominantly $\alpha_{2C}$-adrenoreceptors mediating the vasospasms seen in RP patients[4]. Increased expression of the transcription factor *IRX1* in muscle, and possibly arterial tissue, may further alter genes involved in prostaglandin and/or bradykinin sensing and production, providing a putative second novel mechanism identified in our genetic study. We lastly observed no strong evidence for a shared genetic architecture with suspected cardiovascular comorbidities for primary RP but did observe a novel detrimental effect of low

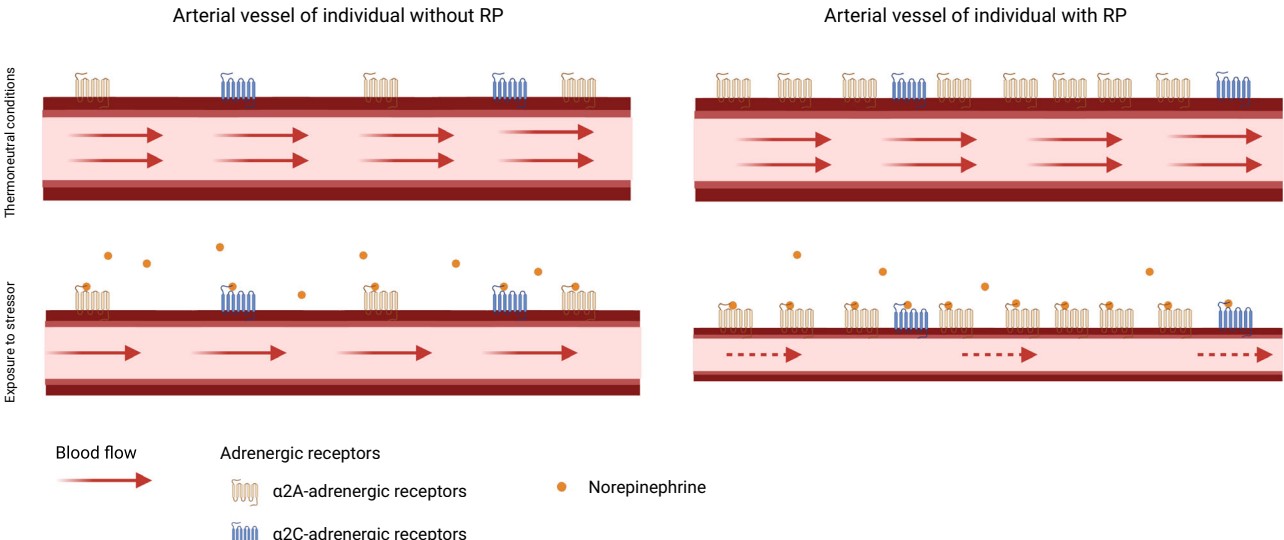

**Fig. 5 | Scheme of the key finding.** In people without Raynaud's phenomenon (RP) stimulation of $\alpha_{2A}$- and $\alpha_{2C}$-adrenoreceptors contributes to vasoconstriction and lower blood flow. In patients with RP, blood flow is further reduced due to overexpression of $\alpha_{2A}$-adrenoreceptors creating a state of hypersensitivity to catecholamine-release. Created with BioRender.com.

fasting plasma glucose that might help to guide primary RP management.

Vasospastic attacks in RP have been repeatedly attributed to cold-induced activation of $\alpha_{2C}$-adrenoreceptors[4,18,33,34], but attacks also occur under thermoneutral conditions and selective, pharmacological blocking of $\alpha_{2C}$-adrenoreceptors did not mitigate symptoms in clinical trials[35]. An observation that aligns with the absence of an effect of a strong cis-eQTL for *ADRA2C* (rs35729104), encoding for $\alpha_{2C}$-adrenoreceptors, on RP risk (OR = 1.02, $p$ = 0.29) in our study. We, in contrast, provide genetically anchored evidence that $\alpha_{2A}$-adrenoreceptors play an underrated role further supported by adverse drug reactions seen with $\alpha_{2A}$-adrenoreceptor agonists (Fig. 5). More specifically, $\alpha_{2A}$-adrenoreceptors mediate arterial vasoconstriction, thrombus stabilization, and hypothermic effects on body temperature[36,37]. Our results are in line with a vasoconstrictive role of $\alpha_{2A}$-adrenoreceptors under thermoneutral conditions[17] and subsequent RP risk and further support findings of a quantitively stronger role of $\alpha_2$-adrenoreceptors compared to $\alpha_1$-adrenoreceptors during cold stress in RP patients[38,39]. However, it might still be conceivable that both $\alpha_2$-adrenoreceptors act in a context-specific manner to induce vasospastic effects[19] or another mechanism partakes in cold-induced RP attacks since pharmacological inhibition of $\alpha_2$-adrenergic receptors did blunt but not eliminate cold-induced vasoconstriction[38].

Currently, RP is treated depending on its severity. For mild cases, avoidance of triggers like cold or emotional stress might be sufficient, but severe vasospastic attacks require pharmacological interventions with calcium channel blockers as first-line response[6]. However, the overall effectiveness of calcium channel blockers[5] or alternative vasodilatory medications such as angiotensin receptor blockers, selective serotonin reuptake inhibitors, or phosphodiesterase-5 inhibitors is limited[6,7]. Negative findings that overall align well with the absence of genetic evidence for those targets in our study. A staggering characteristic of the *ADRA2A* locus was the specificity in phenome-wide screens and comprehensive database lookups, which indicated an RP-specific effect. Such a characteristic may make the inhibition of the gene product, $\alpha_{2A}$-adrenoreceptors, an interesting pharmacological target. This does not rule out that systemically administered antagonists might exert adverse effects, but inhibiting $\alpha_{2A}$-adrenoreceptor activity in disease-relevant tissue, for example using topical solutions, might provide a safe and effective treatment option. There are already approved medications that target $\alpha_2$-adrenoreceptor antagonistically like the widely prescribed antidepressant mirtazapine, but they lack specificity for $\alpha_{2A}$-adrenoreceptor[36] and case reports described the occurrence or worsening of RP in two male patients under yohimbine treatment, another $\alpha_2$-adrenoreceptor antagonist, for erectile dysfunction[40,41]. Clinical trials are needed to test whether treatment with mirtazapine, or topical solutions thereof, can indeed mitigate vasospastic attacks in patients with primary RP, also because an early study described relieved symptoms under methyldopa, an $\alpha_2$-adrenergic agonist[42].

In addition to hypersensitivity to vasoconstrictive stimuli, missing vasodilation is the primary cause of persistent vasospasms in RP. We identified overexpression of *IRX1*, the second strongest signal for RP, in muscle cells and possibly the tibial artery as a putative RP risk-increasing mechanism. While little is known about *IRX1* in general, a transfection study in gastric cancer cell lines identified several putative target genes of *IRX1* with a role in vasoconstriction and vasolidation[43]. This included the downregulation of *PTGER1*, encoding the prostaglandin E2 receptor 1 (EP1) and upregulation of *PTGS2*, encoding prostaglandin G/H synthase 2, as well as downregulation of *BDKRB2*, encoding bradykinin receptor B2. Prostaglandin E2 (PGE$_2$) can induce both vasodilation and vasoconstriction depending on receptor binding, with binding to EP1 inducing vasoconstriction in smooth muscle cells[44,45]. Downregulation of *BDKRB2* in endothelial and smooth muscle cells, in turn, may blunt the vasodilative function of bradykinin[46], in line with observations in RP patients treated with bradykinin[47,48]. A putative role of the *IRX1* locus in vasodilation/vasoconstriction might also explain its association with cardiovascular disease as described among people of East Asian descent[24–26]. Further experimental work will help clarify this, and might also help to refine the application of prostaglandin or bradykinin analogues, as vasodilators in RP[4].

Little is known about the causes of primary RP, and we provided evidence that genetic liability towards low fasting glucose levels increases RP risk. A putative mechanism, however, remains elusive and we did not observe evidence that other metabolic risk factors were causally associated with RP risk, including BMI (Supplementary Data 3). Overexpression of $\alpha_{2A}$-adrenoreceptors in pancreatic islet cells has been postulated as a mechanism for diminished insulin secretion and postprandial hyperglycaemia among carriers of risk

alleles in *ADRA2A*[49]. However, the reported risk allele, rs553668, is only weakly correlated ($r^2 = 0.15$) with our RP lead signal at the same locus and the same applies to the more recently identified genome-wide significant finding[50]. Although likely via distinct genetic variants, both diseases seem to share overexpression of *ADRA2A* as a common, but tissue-specific, disease mechanism, and it is unclear whether over-expression of *ADRA2A* in pancreatic beta-cells also contributes to fasting glucose levels. Conversely, hypoglycaemia exerts profound changes on the vasculature, including vasoconstriction via increased adrenal catecholamine release to maintain blood flow to the brain[51], which may, in turn, exaggerate hypersensitivity to catecholamines of arteriovenous anastomoses in fingers and toes. However, a drop in body temperature following hypoglycaemia is caused by cutaneous vasodilation, which might itself be a trigger for vasospastic effects in RP patients. Even a role of β-adrenergic receptors might be con-ceivable since beta-blockers can cause both, hypoglycaemia[52] and RP[4]. A pragmatic consequence for people at RP risk, or even RP patients, might be to avoid episodes of low plasma glucose levels.

While we identified eight loci robustly associated with RP, we did not find support for genes highlighted by previous smaller-scale and candidate gene studies. This included subthreshold findings from an earlier GWAS[14] (*NOS1*, smallest *p*-value in *NOS1* region ± 500 kb ($p_{min} = 2.6 \times 10^{-3}$) as well as candidate gene studies for serotonin 1B (*HTR1B*; $p_{min} = 6.8 \times 10^{-4}$) and 1E (*HTR1E*; $p_{min} = 1.2 \times 10^{-5}$) receptors and the beta subunit of the muscle acetylcholine receptor (*CHRNB1*; $p_{min} = 1.3 \times 10^{-3})$[13]. Our results therefore highlight the need for large sample sizes, which are only now possible with the increased avail-ability of electronic health record linkage.

Our study has limitations that need to be considered in the interpretation of the results. Firstly, we defined patients with RP based on billing codes from medical records of various sources which might have missed patients with milder presentations and suffer from potential misclassification if patient symptoms have been wrongly assigned to RP, in particular in the presence of disease that causes RP as a secondary phenomenon. However, our conservative sensitivity analysis, excluding all patients with potential known causes of RP revealed congruent findings and the associated loci have a clear bio-logical link towards RP pathology. Secondly, our study was exclusively comprised of participants of European descent and low case numbers and poor imputation quality of the main locus did not permit robust analysis in other ancestries, which is needed to achieve fair repre-sentation of different ancestries in genomic studies and may reveal additional findings. Thirdly, while we were able to replicate the asso-ciation between *ADRA2A* expression in tibial artery and risk of RP in a small, independent cohort ($n = 124$ RP patients; OR = 1.27; $p = 0.0039$) from the BioVU study[53], additional efforts in large biobanks with linked primary care data are needed to further consolidate our findings. Finally, while we assign putative candidate genes at two loci, more work is needed to improve gene assignment pipelines to elucidate possible mechanisms at other loci including rare variants with strong effect sizes.

Our results advance the understanding of RP pathology by shift-ing the focus from a merely cold-induced phenomenon to a clinical entity in its own right with a distinct genetic architecture. The identi-fication of *ADRA2A* and *IRX1* as candidate causal genes will further inform the development and refinement of treatment strategies although more experimental work is needed.

## Methods

### United Kingdom Biobank (UKBB)

UK Biobank is a prospective cohort study from the UK that contains more than 500,000 volunteers between 40 and 69 years of age at inclusion. The study design, sample characteristics, and genome-wide genotype data have been described in Sudow et al. and Bycroft et al.[15,54]. The UKBB was approved by the National Research Ethics

Service Committee North West Multi-Centre Haydock and all study procedures were performed in accordance with the World Medical Association Declaration of Helsinki ethical principles for medical research. We included 444,441 individuals in the GWAS for whom inclusion criteria (given consent to further usage of the data, avail-ability of genetic data, and passed quality control of genetic data) applied. Data from the UKBB were linked to death registries, hospital episode statistics (HES), and primary care data.

### RP case ascertainment

We used all available electronic health records, including primary (~45% of the population) and secondary care, death certificates as well as participant's self-report to define RP cases. We manually curated relevant codes for RP based on the different coding systems used (ICD-10 codes: I73.0, I73.00, I73.01; CTV3/Read2: G730., G7300, G7301, G730z, XE0VQ) and flagged the first occurrence of a related code in the records of a participant as the incidence date. We compared the date of the very first record related to RP with the date of the baseline examination of the same participant to distinguish between prevalent and incident RP. We followed the criteria from Wigley et al.[4] to dif-ferentiate between primary and secondary RP. In detail, we classified each RP case as 'secondary' when patients were either under beta-blocker treatment or had been diagnosed with one of the following diseases prior or at the same time of their RP diagnosis: arterio-sclerosis, carpal tunnel syndrome, systemic lupus erythematosus, systemic sclerosis, polymyositis, dermatomyositis, Sjögren syndrome, hypothyroidism, cryoglobulinemia, frostbite or other connective tis-sues diseases (Supplementary Data 6).

### UKBB medication classification/ beta-blocker usage

During the baseline assessment self-reported regular medication use was recorded. Medication data were categorized into 6745 groups (Data field: 20,003). 38 categories of the medication data were iden-tified manually to represent medication containing beta-blocker substances.

### Demographic analysis

We performed $\chi^2$-tests and analysis of variance to test for statistically significant differences in demographic characteristics between con-trols and RP cases using standard implementations in R (v4.1.2).

### Genotyping, quality control and participant selection

Details on genotyping for UKBB have been reported in detail by Bycroft et al.[15]. Briefly, we used data from the 'v3' release of UKBB containing the full set of Haplotype Reference Consortium (HRC) and 1000 Genomes imputed variants. We applied recommended sample exclusions by UKBB including low-quality control values, sex mis-match, and heterozygosity outliers. We defined a subset of 'white European' ancestry by clustering participants based on the first four genetic principal components derived from the genotyped data using a k-means clustering approach with k = 5. We classified all participants who belonged to the largest cluster and self-identified as of being 'white,' 'British', 'Any other white background', or 'Irish' as 'white Eur-opean'. After the application of quality control criteria and dropping participants who had withdrawn their consent, a total of 444,441 UKBB participants were available for analysis with genotype and phenotype data.

We used only called or imputed genotypes and short insertions/deletions (here commonly referred to as SNPs for simplicity) with a minor allele frequency (MAF) > 0.001%, imputation score > 0.4 for common (MAF ≥ 0.5%) and >0.9 for rare (MAF < 0.5%), within Hardy–Weinberg equilibrium ($p_{HWE} > 10^{-15}$), and minor allele count (MAC) > 10. This left us with 15,519,342 autosomal and X-chromosomal variants for statistical analysis. GRCh37 was used as reference genome assembly.

## Genome-wide association study

We performed genome-wide association studies for RP and primary RP using REGENIE v2.2.4 via a two-step procedure to account for population structure as described in detail elsewhere[55]. We used a set of high-quality genotyped variants (MAF > 1%, MAC > 100, missingness < 10%, $p_{HWE}$ > $10^{-15}$) in the first step for individual trait predictions using the leave one chromosome out (LOCO) scheme. These predictions were used in the second step as offset to run logistic regression models with saddle point approximation to account for case/control imbalance and rare variant associations. Models were adjusted for age, sex, genotyping batch, assessment centre, and the first ten genetic principal components. We used a tier system to classify genome-wide significant loci. Given the absence of a substantial replication cohort, we consider loci meeting a stringent p-value threshold of $p < 1.0 \times 10^{-9}$ as robust findings (tier 1) and further considered loci meeting the standard genome-wide significance threshold of $p < 5.0 \times 10^{-8}$ as strongly suggestive loci (tier 2).

## LD score regression and genetic correlation to other common diseases

We tested for genomic inflation and calculated the SNP-based heritability using LD-score regression (LDSC v1.0.1)[56]. We further used LDSC to compute the genetic correlation between RP as well as primary RP and a set of 20 preselected traits and diseases (Supplementary Data 2). *P*-values were corrected for multiple testing by controlling the false-discovery rate at 5% according to the Benjamini–Hochberg procedure. To explore genetic correlation with other common diseases, we computed genetic correlation to a set of 185 phecodes (see below; Supplementary Data 4) with at least 1% SNP-based heritability derived from the UKBB.

## Signal selection and fine-mapping

We used regional clumping (±500 kb) to select independent genomic regions associated with RP treating the extended MHC region as a whole (chr6:25.5-34.0 Mb) and collapsing neighbouring regions using BEDtools v1.5.

Within each region, apart from the MHC region, we performed statistical fine-mapping using "Sum of single effects" model (SuSiE)[57] as implemented in the R package *susieR* (v.0.11.92). Briefly, SuSiE employs a Bayesian framework for variable selection in a multiple regression problem with the aim to identify sets of independent variants each of which likely contains the true causally underlying genetic variant. We implemented the workflow using default prior and parameter settings, apart from the minimum absolute correlation, which we set to 0.1. Since SuSiE is implemented in a linear regression framework, we used the GWAS summary statistics with a matching correlation matrix of dosage genotypes instead of individual-level data to implement fine-mapping (*susie_rss()*) as recommended by the authors[57]. We computed the correlation matrix with LDstore v1.0.

## Sex-specific SNP effect

We tested for a potential modulating effect of sex on SNP–RP association using an additional interaction term in logistic regression models with the same adjustment set as described above. We implemented this analysis in R v.4.1.2 using a set of 361,781 unrelated white European individuals.

## Replication in East London Genes & Health

We performed a genome-wide association analysis of RP in the Genes & Health cohort of British individuals of South Asian ancestry. We defined RP status using electronic health records and using the same code list as for the European-ancestry cohort. We identified 401 RP cases and 40,131 controls with complete covariate and genetic data passing quality control[30] (mean age at recruitment: 41.2 yr [s.d.*D*:14.2], 55.6% females).

Genotyping and quality control for Genes & Health is described in Supplemental File 1. Briefly, genotyping was done using the Illumina Infinium Global Screening Array v3 with additional multi-disease variants and imputed to TopMed version 3 using the Michigan Imputation Server. We used REGENIE[55] for association testing with age, sex, age[2], and the first twenty genetic principal components as covariates. We included only autosomal genetic variants with an MAF > 1% (in this cohort), INFO score > 0.7 for imputed variants, $p_{HWE}$ > $10^{-10}$, and missingness < 10%. To allow for comparison with the results from the UK Biobank, we used the liftOver tool (v.2.0) to map between hg19 and hg38.

## eQTL mapping to tissues/functional annotation

We systematically tested for a shared genetic signal between RP loci and gene expression levels (eQTL) in 49 tissues from the GTEx project (v8)[22]. Briefly, we considered all protein-coding genes or processed transcripts encoded in a 1 Mb window around RP loci and performed statistical colocalization[58] to test for a shared genetic signal between RP and gene expression, as implemented in the R *coloc* package (v.2.0). We only report RP loci–gene–tissue triplets with a posterior probability >80% for a shared genetic signal as candidate causal genes. We adopted a recently recommended prior setting[59] with $p_{12} = 5 \times 10^{-6}$. All GTEx variant-gene cis-eQTL associations from each tissue were downloaded in January 2020 from https://console.cloud.google.com/storage/browser/gtex-resources.

We further used the PrediXcan software[60] to associate genetically predicted expression levels of *ADRA2A* with the risk of RP in 124 RP cases and 16,965 controls of the BioVU cohort[53].

## Phenome-wide association studies and variant look-up

We performed phenome-wide association studies for each RP locus identified using the comprehensive electronic health record linkage to generate a set of 1448 'phecodes', from which we used 1155 'phecodes' with a case number of more than 200. To generate phecode-based outcome variables, we mapped ICD-10, ICD-9, Read version 2, Clinical Terms Version 3 (CTV3) terms codes from self-report or medical health records, including cancer registry, death registry, hospitalization (e.g., Hospital Episode Statistics for England), and primary care, to a set of summarized clinical entities called phecodes[61,62]. For example, more than 90 ICD-10 codes can indicate participants with type 1 diabetes that are here collectively summarized under the phecode 'type 1 diabetes'[63]. We used any code that was recorded, irrespective if it contributed to the primary cause of death or hospital admission, to define phecodes. We adjusted all analyses for the test centre to account for regional differences in coding systems and case ascertainment. For each participant and phecode, we kept only the first entry irrespective of the original data set, generating a first occurrence data set. We dropped codes that were before or in the participants' birth year to minimize coding errors from electronic health records. We implemented the same logistic regression framework used for sex-interaction testing among unrelated individuals of White European ancestry. We applied a stringent Bonferroni correction to account for multiple testing ($p < 3.7 \times 10^{-5}$).

We further performed look-ups for all independent lead variants and proxies ($r^2 > 0.6$) associated with RP in the OpenTargets database[21] and the GWAS catalogue[20] (download: 27/04/2022) to search for possibly previously reported traits and to establish novelty.

## LCV analysis

To assess the evidence for a causal relationship between RP and each of the genetically correlated phenotypes (Supplementary Data 3 and 4) we implemented a latent causal variable (LCV) model[31]. The LCV model estimates a parameter termed genetic causality proportion (GCP). The values of GCP range from −1 to 1, quantifying both the magnitude and direction of genetic causality between two traits. Briefly, the LCV model fits an unobserved latent variable, *L*, which mediates the genetic

correlation between two traits and compares the correlation of each trait with *L* to estimate the GCP. A|GCP| value of one indicates full genetic causality (suggestive of vertical pleiotropy) whereas a GCP value of zero implies no evidence for genetic causality (suggesting that the genetic correlation between two traits is likely to be mediated by horizontal pleiotropy or shared risk factors) and |GCP| < 1 indicates partial genetic causality between two traits. Negative values of the GCP estimate suggest that RP lies downstream of the other trait and interventions on the other trait are likely to affect RP, while positive GCP values indicate that the other trait lies downstream of RP. We implemented LCV analysis using standard parameters with version 1.0. We used the same LD-scores as in the genetic correlation analysis and excluded the extended MHC regions from any calculations. We considered the number of significant genetic correlations as a testing burden for LCV results.

### Drug target annotation

To identify putative druggable targets for RP and test for evidence for drugs already in clinical trials, we queried the Open Target database[21] for all genes likely linked to GWAS loci (i.e., evidence from eQTL annotation or intronic). We augmented this gene list by functionally validated interaction partners from the SIGNOR network[32] and retained all matching drugs for further investigation. We lastly manually screened all prioritized drugs to match mechanisms to signs and symptoms of RP.

### Reporting summary

Further information on research design is available in the Nature Portfolio Reporting Summary linked to this article.

## Data availability

All individual-level data is publicly available to bona fide researchers from the UK Biobank (https://www.ukbiobank.ac.uk/) or Genes & Health study (https://www.genesandhealth.org/). GWAS summary statistics have been deposited in the GWAS Catalogue (accession codes: GCST90271713 and GCST90271714). Lookups and druggable targets were derived from the GWAS Catalogue https://www.ebi.ac.uk/gwas/ and the OpenTargets Genetics data base (https://genetics.opentargets.org/).

## Code availability

Associated code and scripts for the analysis are available on Github[64] (https://github.com/pietznerm/gwas_rp).

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

## Acknowledgements

All research related to UK Biobank has been done under the application number 44448. Mu. P. wishes to thank the MRC (MR/L006758/1, MR/V033867/1) and NIHR (NIHR202349) for their support. This collaboration is part of the National Institute For Health Barts Biomedical Research Centre portfolio. B.M.J. is funded by a Medical Research Council (MRC) Clinical Research Training Fellowship (CRTF) jointly supported by the UK MS Society (B.M.J.; grant reference MR/V028766/1), and by Barts Charity. The authors acknowledge the Scientific Computing of the IT Division at the Charité—Universitätsmedizin Berlin for providing computational resources that have contributed to the research results reported in this paper (https://www.charite.de/en/research/research_support_services/research_infrastructure/science_it/#c30646061). Genes & Health is/has recently been core-funded by Wellcome (WT102627, WT210561), the Medical Research Council (UK) (M009017, MR/X009777/1), Higher Education Funding Council for England Catalyst, Barts Charity (845/1796), Health Data Research UK (for London substantive site), and research delivery support from the NHS National Institute for Health Research Clinical Research Network (North Thames). Genes & Health is/has recently been funded by Alnylam Pharmaceuticals, Genomics PLC; and a Life Sciences Industry Consortium of Astra Zeneca PLC, Bristol-Myers Squibb Company, GlaxoSmithKline Research and Development Limited, Maze Therapeutics Inc, Merck Sharp & Dohme LLC, Novo Nordisk A/S, Pfizer Inc, Takeda Development Centre Americas Inc. We thank Social Action for Health, Centre of The Cell, members of our Community Advisory Group, and staff who have recruited and collected data from volunteers. We thank the NIHR National Biosample Centre (UK Biocentre), the Social Genetic & Developmental Psychiatry Centre (King's College London), Wellcome Sanger Institute, and Broad Institute for sample

processing, genotyping, sequencing and variant annotation. We thank: Barts Health NHS Trust, NHS Clinical Commissioning Groups (City and Hackney, Waltham Forest, Tower Hamlets, Newham, Redbridge, Havering, Barking and Dagenham), East London NHS Foundation Trust, Bradford Teaching Hospitals NHS Foundation Trust, Public Health England (especially David Wyllie), Discovery Data Service/Endeavour Health Charitable Trust (especially David Stables), NHS Digital—for GDPR-compliant data sharing backed by individual written informed consent. Most of all we thank all of the volunteers participating in Genes & Health.

## Author contributions

Conceptualization: Ma. P. and C.L. Data curation/Software: S.H., S.Y., B.M.J., S.D. and Ma. P. Formal analysis: S.H., S.Y., B.M.J., E.G. and Ma. P. Methodology: S.D. and Ma. P. Visualization: S.H., S.Y. and Ma. P. Funding acquisition: C.L. and H.H. Project administration: C.L. and H.H. Supervision: Ma. P. and C.L. Writing—original draft: S.H., Ma. P. and C.L. Writing—review & editing: S.Y., B.M.J., S.D., H.H., Mu. P. and M.C.

## Funding

## Competing interests

Mu. P. has received partnership funding for the following: MRC Clinical Pharmacology Training Scheme (co-funded by MRC and Roche, UCB, Eli Lilly, and Novartis); and a Ph.D. studentship jointly funded by EPSRC and Astra Zeneca. Mu. P. also has unrestricted educational grant support for the UK Pharmacogenetics and Stratified Medicine Network from Bristol-Myers Squibb. Mu. P. has developed an HLA genotyping panel with MC Diagnostics, but does not benefit financially from this. He is part of the IMI Consortium ARDAT (www.ardat.org). None of the funding outlined above is related to the current paper. None of the other authors have competing interests.

## Additional information

## Genes & Health Research Team

**Benjamin M. Jacobs** ⓘ [2]

