## [Peer Review File · Nature Communications]

ADRA2A and IRX1 are putative risk genes for Raynaud's phenomenonREVIEWER COMMENTS

Reviewer #1 (Remarks to the Author):

This study is the largest GWAS about a common vasospastic disorder RP. The sample size, statistics and correlation analyses with gene expression and clinical phenome are sufficient and sound. However, it's at most a correlation study far from disease-causing, and all the identified loci are intergenic. Therefore, it's better to be prudent when mentioning about the associated genes throughout the manuscript, including the title.

(1) The conclusion in the abstract goes a bit too far. This study has discovered a role of ADRA2A in the risk of RP. It could only add α 2A-adrenoreceptor signalling as a new unrealized factor, but could not support the conclusion of challenging the current predominance of α 2C adrenoreceptors.

(2) In fact, not all the closest genes listed in Table 1 are the true closest genes illustrated in Supplementary Figure 2, but the closest protein-coding genes. It should be described clearly.

(3) It's better to show the raw p-values besides *fdr* in Supplementary Table 5b.

(4) It seems that all p-values shown in the manuscript Line 220-230 are FDR adjusted p, but I could not find what the p-value for fasting glucose was, not the two *fdrs* listed in Supplementary Table 5b.

(5) All the genetic correlations noted in the manuscript are analyses of all RP cases, except fasting glucose, which is analysis of primary RP cases. This should be consistent.

(6) I'm confused about the genetic correlations with HDL-cholesterol levels and total triglyceride levels. Their directions of correlations written in the manuscript are opposite to those listed in Supplementary Table 5b and Supplementary Fig. 5. Please check.

(7) The intergenic genome-wide significant locus is > 20kb away from ADRA2A, together with the evidence of increased expression of α 2A-adrenergic receptors, are all correlative evidence, thus a bit far from likely disease-causing. It's better to be cautious when using the term 'disease-causing' in the Result section, especially in subtitles. Similarly, the study only observed correlation of the locus with expression of α 2A-adrenoreceptors in arterial tissue, which could not lead to the statement that the locus acts on increasing expression of α 2A-adrenoreceptors. Whether the locus would affect any gene expression need more experiments.

(8) Is there anything missing or typo in Line 379 between 'entity' and 'it'?

Reviewer #2 (Remarks to the Author):

This paper reports a single-stage GWAS for Raynaud's phenomenon in a white European sub-group of the UK Biobank cohort. It is well-written and coherent. The methodology is sound (excepting the major concern below), and the analyses are carefully conducted. The reporting of the methodology is clear and comprehensive, meets the expected standards in the field, and could be easily reproduced.

The follow-on analyses after GWAS are appropriate and comprehensive. The overall message of the paper is novel, especially relating to Alpha 2A adrenoreceptors, and of considerable interest to the scientific and medical community. It may lead to the development of new treatments or drug repurposing opportunities for patients with this common condition.

Major concern

My only concern with this study is the lack of an independent replication cohort. The attempted replication in Genes and Health was a) underpowered, and b) inadequate (the top SNP was not available and had no proxy). I note that the authors have gone to great lengths to make sure that they do not over claim results of statistical significance, and have only followed up on the loci where there is the most evidence of association. However, the lack of even nominal significance of the one tested SNP in Genes and Health casts some doubt on the validity of the results. There is no easy solution for this problem - replication will require collaboration with another biobank that has adequately coded cases of RP, and I do not know if this is available in any of the biobanks worldwide.

It will be an editorial decision as to whether or not to accept a paper with a flaw such as this. On balance, given the quality of the genetic methodology, and the careful follow up work, together with convincing, consistent biological mechanistic interpretation, I would be in favour of acceptance. If the paper is accepted, a more thorough discussion and highlighting of this limitation is warranted.

REVIEWER COMMENTS

Reviewer #1 (Remarks to the Author):

This study is the largest GWAS about a common vasospastic disorder RP. The sample size, statistics and correlation analyses with gene expression and clinical phenome are sufficient and sound. However, it's at most a correlation study far from disease-causing, and all the identified loci are intergenic. Therefore, it's better to be prudent when mentioning about the associated genes throughout the manuscript, including the title.

We thank the reviewer for her/his positive feedback and revised the entire manuscript to be clearer about what level of evidence we provide that associates ADRA2A and other genes with the etiology of RP.

(1) The conclusion in the abstract goes a bit too far. This study has discovered a role of ADRA2A in the risk of RP. It could only add α 2A-adrenoreceptor signalling as a new unrealized factor, but could not support the conclusion of challenging the current predominance of α 2C adrenoreceptors.

We agree with the reviewer that the previous conclusion in the abstract has been too strong and rephrased it accordingly (p2, lines 37-39).

(2) In fact, not all the closest genes listed in Table 1 are the true closest genes illustrated in Supplementary Figure 2, but the closest protein-coding genes. It should be described clearly.

We followed this helpful recommendation and clarified the legend of the table.

(3) It's better to show the raw p-values besides fdr in Supplementary Table 5b.

We now provide raw p-values along with FDR values in Supplemental Table 5b.

(4) It seems that all p-values shown in the manuscript Line 220-230 are FDR adjusted p, but I could not find what the p-value for fasting glucose was, not the two fdrs listed in Supplementary Table 5b.

We apologize for the missing information and now provide raw p-values in the text along with FDR values in Supplementary Tables (pages 6/7, lines 169-178).

(5) All the genetic correlations noted in the manuscript are analyses of all RP cases, except fasting glucose, which is analysis of primary RP cases. This should be consistent.

We apologize for the unclear presentation in the previous version of the manuscript. We now clarify that genetic correlations have been run for 1) all RP cases and 2) primary RP cases only, but only significant findings from genetic correlation analysis were taken forward for latent causal variable analysis (p6/7, lines 169-178).

(6) I'm confused about the genetic correlations with HDL-cholesterol levels and total triglyceride levels. Their directions of correlations written in the manuscript are opposite to those listed in Supplementary Table 5b and Supplementary Fig. 5. Please check.

We apologize for the confusion and thank the reviewer for highlighting this important typo that has now been corrected in the revised version of the manuscript.

(7) The intergenic genome-wide significant locus is > 20kb away from ADRA2A, together with the evidence of increased expression of α_2A -adrenergic receptors, are all correlative evidence, thus a bit far from likely disease-causing. It's better to be cautious when using the term 'disease-causing' in the Result section, especially in subtitles. Similarly, the study only observed correlation of the locus with expression of α_2A -adrenoreceptors in arterial tissue, which could not lead to the statement that the locus acts on increasing expression of α_2A -adrenoreceptors. Whether the locus would affect any gene expression need more experiments.

We agree with the reviewer that linking common, intergenic variants to disease-causing genes is challenging and we revised section headers and wording in the manuscript, accordingly, referring to 'likely effector genes' instead to highlight the remaining uncertainty.

However, we provide evidence from different sources that converge on ADRA2A as the most likely candidate gene at this locus: 1) common human sequence variation at the locus strongly associates with RP risk, irrespective of biological sex and clinical definition, 2) the exact same genetic signal (rs7090046; PP=96%%) increases ADRA2A gene expression in disease relevant tissue, tibial artery, based on the GTEx resource, 3) medications stimulating α_2A -adrenoreceptors, such as clonidine, are known to cause RP, and 4) no other gene within 2Mb window of our main signal has a similar obvious pathological link to the vasospasms characterizing RP.

While these findings certainly warrant further experimental follow-up, in particular genotype-dependent α_2A -adrenergic receptor expression at the protein level, as the reviewer rightly outlines, we believe that the cumulative evidence strongly implies ADRA2A as the effector gene at this locus.

(8) Is there anything missing or typo in Line 379 between 'entity' and 'it'?

We thank the reviewer for pointing out this typo that has now been corrected in the revised version of the manuscript.

Reviewer #2 (Remarks to the Author):

This paper reports a single-stage GWAS for Raynaud's phenomenon in a white European subgroup of the UK Biobank cohort. It is well-written and coherent. The methodology is sound (excepting the major concern below), and the analyses are carefully conducted. The reporting of the methodology is clear and comprehensive, meets the expected standards in the field, and could be easily reproduced.

The follow-on analyses after GWAS are appropriate and comprehensive. The overall message of the paper is novel, especially relating to Alpha 2A adrenoreceptors, and of considerable interest to the scientific and medical community. It may lead to the development of new treatments or drug repurposing opportunities for patients with this common condition.

We thank the reviewer for his/her tremendously positive feedback on our work.

Major concern

My only concern with this study is the lack of an independent replication cohort. The attempted replication in Genes and Health was a) underpowered, and b) inadequate (the top SNP was not available and had no proxy). I note that the authors have gone to great lengths to make sure that they do not over claim results of statistical significance, and have only followed up on the loci where there is the most evidence of association. However, the lack of even nominal significance of the one tested SNP in Genes and Health casts some doubt on the validity of the results. There is no easy solution for this problem - replication will require collaboration with another biobank that has adequately coded cases of RP, and I do not know if this is available in any of the biobanks worldwide.

We agree with the reviewer that the replication in Genes and Health was not ideal and now clearly discuss this limitation in the manuscript (pages 10/11, lines 307-310).

The reviewer also rightly points out the difficulties for replication, since RP is mostly diagnosed in primary care, linkage to which is absent in most biobanks, and poorly captured by self-report. However, we now provide now at least replication at the gene expression level for our main finding based on 124 RP patients and 16,965 controls from the BioVU study (OR=1.27; $p = 0.0039$; page 10, lines 307-308), but clearly emphasize that more effort to replicate our findings are needed.

It will be an editorial decision as to whether or not to accept a paper with a flaw such as this. On balance, given the quality of the genetic methodology, and the careful follow up work, together with convincing, consistent biological mechanistic interpretation, I would be in favour of acceptance. If the paper is accepted, a more thorough discussion and highlighting of this limitation is warranted.

We thank the reviewer for his/her support of our work and followed the recommendation to discuss the limitations in our replication approach more thoroughly (p10/11, lines 305-310).

REVIEWERS' COMMENTS

Reviewer #1 (Remarks to the Author):

The manuscript has been revised to be fine in statement accuracy. However, I have one more suggestion. Since the authors have a small replication cohort BioVU study, why not show the replication results for all the major findings in the Results section, including in the tables?

Reviewer #2 (Remarks to the Author):

The authors have addressed all of the concerns raised in the peer review of the original manuscript. They have tempered their conclusions, made an attempt at a form of replication, and discussed limitations more thoroughly. I recommend publication.

REVIEWERS' COMMENTS

Reviewer #1 (Remarks to the Author):

The manuscript has been revised to be fine in statement accuracy. However, I have one more suggestion. Since the authors have a small replication cohort BioVU study, why not show the replication results for all the major findings in the Results section, including in the tables?

We appreciate the suggestion of the reviewer and agree that transparent reporting of replication results should be provided whenever possible. However, our replication is unfortunately limited to a study ninety times smaller than our discovery study, with only sufficient power to replicate the strongest finding. In other words, reporting negative findings for other loci with overlapping eQTL findings in BioVU, may wrongly lead to the conclusion that those loci are not supported.

We clearly acknowledge these limitations in the discussion section of the manuscript, and only report replication of the ADRA2A findings in this context. We therefore hope that the editor agrees, that reporting our replication results as part of the main findings in a table would be an exaggeration of the underlying evidence.

We note that obtaining the replication for ADRA2A already took weeks, and we anticipate that obtaining, likely underpowered, results for other genes would take even longer with unlikely gain in the coherence of the presented findings.

Reviewer #2 (Remarks to the Author):

The authors have addressed all of the concerns raised in the peer review of the original manuscript. They have tempered their conclusions, made an attempt at a form of replication, and discussed limitations more thoroughly. I recommend publication.

We thank the reviewer for the positive assessment of our work.